# Social environment mediates cancer progression in *Drosophila*

Erika H. Dawson [1,2], Tiphaine P. M. Bailly[1], Julie Dos Santos[1], Céline Moreno[1], Maëlle Devilliers[3], Brigitte Maroni[3], Cédric Sueur [4,5], Andreu Casali[6], Beata Ujvari [7], Frederic Thomas[8], Jacques Montagne[3] & Frederic Mery[1]

The influence of oncogenic phenomena on the ecology and evolution of animal species is becoming an important research topic. Similar to host–pathogen interactions, cancer negatively affects host fitness, which should lead to the selection of host control mechanisms, including behavioral traits that best minimize the proliferation of malignant cells. Social behavior is suggested to influence tumor progression. While the ecological benefits of sociality in gregarious species are widely acknowledged, only limited data are available on the role of the social environment on cancer progression. Here, we exposed adult *Drosophila*, with colorectal-like tumors, to different social environments. We show how subtle variations in social structure have dramatic effects on the progression of tumor growth. Finally, we reveal that flies can discriminate between individuals at different stages of tumor development and selectively choose their social environment accordingly. Our study demonstrates the reciprocal links between cancer and social interactions and how sociality may impact health and fitness in animals and its potential implications for disease ecology.

[1] Evolution, Génomes, Comportement & Ecologie, CNRS, IRD, Université Paris-Sud, Université Paris-Saclay, 91198 Gif-sur-Yvette, France. [2] Unité mixte internationale de Modélisation Mathématique et Informatique des Systèmes Complexes. (UMI IRD/ Sorbonne Université, UMMISCO), 32 Avenue Henri Varagnat, 93143 Bondy Cedex, France. [3] Institut for Integrative Biology of the Cell (I2BC), CNRS, Université Paris-Sud, CEA, UMR 9198, 91190 Gif-sur-Yvette, France. [4] Département Ecologie, Physiologie et Ethologie, Centre National de la Recherche Scientifique, 67037 Strasbourg, France. [5] Institut Pluridisciplinaire Hubert Curien, Université de Strasbourg, 67037 Strasbourg, France. [6] Institut de Recerca Biomèdica de Lleida Fundació Dr. Pifarré (IRBLleida), 25198 Lleida, Spain. [7] Centre for Integrative Ecology, School of Life and Environmental Sciences, Deakin University, Waurn Ponds 3216, Australia. [8] CREEC, MIVEGEC, UMR IRD/CNRS/UM 5290, 34394 Montpellier, France. These authors contributed equally: Erika H. Dawson, Tiphaine P. M. Bailly, Julie Dos Santos, Frederic Thomas, Jacques Montagne, Frederic Mery. Correspondence and requests for materials should be addressed to F.T. (email: frederic.thomas2@ird.fr) or to J.M. (email: Jacques.montagne@i2bc.paris-saclay.fr) or to F.M. (email: frederic.mery@egce.cnrs-gif.fr)

I n gregarious species, sociality not only offers important positive benefits associated with reducing predation risk[1] and increasing foraging efficiency[2], but also provides additional adaptive benefits by reducing overall metabolic demand[3], providing thermal advantages[4], decreasing stress responses[5] and increasing disease avoidance[6]. It is therefore, generally accepted that an individual's social environment affects a large range of behavioral, psychosocial, and physiological pathways. Limited empirical evidence suggests that extreme social environments such as complete isolation or overcrowding of conspecifics in a group can potentially induce and accelerate pathological disorders. For example, in mammals, social isolation has been associated with faster progression of type 2 diabetes[7], cardiovascular or cerebrovascular disorders[8], and, notably, early and faster mammary cancer development. Moreover, social overcrowding has been found to induce psychiatric and metabolic disorders[9]. Few human studies have attempted to explore the role of social interactions on cancer progression (though see ref. [10,11] for non-human animal studies), and the topic remains controversial. Adverse psycho-social factors, including traumatic life events, high levels of depressive symptoms, or low levels of social support, have been related to higher rates of, for example, breast and colon cancers[12,13]. However, these community based studies or meta-analyses often suffer from the complexity of inter-correlated factors. For example, low sample sizes, high risk behaviors associated with stress (e.g., smoking), and the heterogeneity and retrospective origins of these studies make it difficult to find a conclusive causal relationship between cancer progression and social conditions.

Increasing evidence demonstrates that oncogenic phenomena are extremely prevalent in host populations, and not just in post-reproductive individuals as previously believed[14]. While cancer is generally viewed as a senescence-related malady, it also exists at sub-clinical levels in humans and other animals[15]. Even at early stages, tumors will impose a heavy burden on the body[16] (e.g., through tolerance mechanisms), which will undoubtedly have indirect fitness consequences (such as vulnerability to predation), and as a result is likely to be a strong selective force from early on in the lifetime of an organism. Despite cancer (both transmissible and non-transmissible) being an emerging important factor influencing life history traits, even early in life[17–19], little is known regarding the reciprocal links between the social environment and the development and progression of this illness.

*Drosophila* has the potential to be a powerful model system to address the relationship between social group composition and tumor progression. Social interactions are an important life history trait, particularly in female flies who use social information to make fitness enhancing decisions[20–22]. More importantly, behavioral and physiological processes are influenced by social interactions. In *Drosophila*, social isolation leads to a reduced lifespan[23], an increase in aggression[24–26], a reduced need for sleep[27,28], and a decrease in the number of fibers in the mushroom bodies, a center for integration of information in the fly brain[29]. Finally, tumor-like over-proliferation of tissues occurs naturally in *Drosophila*[30,31] and induced tumors influence fitness traits in individuals[19].

Here, we use an established colorectal-like tumor model[32] to explore the reciprocal relationship between social environment and cancer progression. Using genetic tools available for *D. melanogaster*, tumors can be induced during a precise adult developmental stage and subsequently followed over the lifespan of the fly. The tumors are generated by inducing clones in intestinal progenitor cells that are homozygous mutants for the two *Drosophila adenomatous polyposis coli* (*APC*) genes and that express an oncogenic form of the proto-oncogene *Ras*. Loss-of-function of the *APC* tumor suppressor and expression of

oncogenic *Ras* are critical steps towards malignancy in the human colorectal tract[33]. In the present study, we first exposed tumor-bearing *Drosophila* females to various social environments for 21 days and measured tumor progression and social interactions. We then tested the hypothesis that flies with cancer should choose social environments that limit cancer progression. Flies kept in isolation exhibit faster tumor progression than flies kept in homogeneous groups. More importantly, we also found that cancerous flies, kept in homogeneous groups, develop tumors at a lower rate compared to heterogeneous groups, where a single cancerous fly was kept with other non-cancerous conspecifics, suggesting a strong impact of social group composition on cancer growth. Finally, we show that flies can discriminate between individuals at different stages of tumor development and selectively choose their social environment accordingly. These findings highlight the importance of the relationship between social interactions and the development of tumor growth, which may consequently affect the evolutionary ecology of non transmissible diseases.

## Results

**Biological model**. Flies bearing heat shock (HS)-induced MARCM (Mosaic analysis with a repressible cell marker) clones[34], induced in 3-day old adult virgin females intestinal progenitor cells, were used. The clones were mutant for both *Drosophila APC* genes, *Apc* and *Apc2*, and expressed the oncogenic form of *Ras*, $Ras^{V12}$ and the *GFP* marker (Apc-Ras clones)[32]. These compound *Apc-Ras* clones, but not clones expressing either $Ras^{V12}$ or mutated for the *APC* genes alone, expand as aggressive intestinal tumor-like overgrowths that reproduce many hallmarks of human colorectal cancer[32]. One and two weeks after induction of the MARCM recombination, GFP-positive cells were dispersed along the midgut (Supplementary Fig. 1A-B), while 3 weeks after recombination, GFP-positive cells were condensed mostly in one single group in the anterior midgut (Supplementary Fig. 1C) or in the Malpighian tubules. The frontier between groups of tumoral cells and the surrounding control cells was however, difficult to precisely delineate (Supplementary Fig. 1D). Conversely, and as previously shown[32], neutral clones were always dispersed along the midgut at any time after HS induced recombination. Thus, the number of GFP-positive gut cells was monitored over time every 7-day by flow cytometry from flies bearing either *Apc-Ras* or neutral clones (Supplementary Fig. 2), hereafter referred to as cancerous and control flies, respectively. In accordance with the images of dissected guts (Supplementary Fig. 1A-C), a clear increase in the number of GFP-positive tumor cells was observed 3 weeks after clone induction (Supplementary Fig. 2J (ANOVA) $F_{1,33} = 8.6$; $P = 0.006$). The presence of tumor cells (Apc-Ras clones) had little impact on fly performance and survival over the 3 weeks of the experimental study[32] (Supplementary note 1; Supplementary Fig. 4).

**Cancer progression and social environment**. To investigate the impact of the social environment on tumor progression, we exposed adult cancerous females for 21 days, post-induction, to various social environments in 40 ml food tubes. Individual virgin cancerous females were either kept in tubes alone (social isolation), in groups composed of seven other female cancerous flies (homogeneous groups) or in groups with seven non-cancerous control females (heterogeneous groups). Tumor progression was significantly affected by the social environment (Wald $\chi^2_2 = 6.7$, $P = 0.031$): after 21 days we observed that tumor progression was markedly higher in cancerous flies kept in isolation than in cancerous flies kept in homogeneous groups (Fig. 1). More surprisingly, we also observed that cancerous individuals kept within

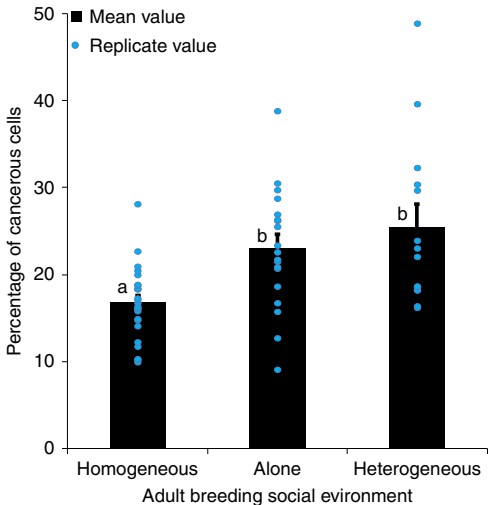

**Fig. 1** Gut tumor progression as a function of social environment. FACS analysis of GFP-positive cells in guts dissected from 21 days post-HS cancerous females as a function of social environment. Blue dots indicate mean value for each replicate. Error bars: standard error of the mean. $N = 15$ measures for each treatment. Letters are Tukey's post-hoc classification

a group of control flies showed an increased number of tumor cells compared to cancerous flies grouped together (Fig. 1).

**Social interactions**. We then analyzed how social interactions were affected by tumor progression and group composition. Using a video tracking setup, we followed the locomotion and interactions of groups of flies (3 weeks post-induction) placed in an arena for 1 h. For social interaction measures we used homogeneous groups of eight control or eight cancerous flies, and a heterogeneous group consisting of seven control and one cancerous fly which were kept together for 21 days post-induction. Social interaction analyses confirmed that control and cancerous flies had similar locomotor activity, independent of their social environment ((ANOVA) Fig. 2a; log (trail length): group composition: $F_{1,84} = 2.64$, $P = 0.1$; fly state: $F_{1,84} = 0.13$, $P = 0.7$; fly state × group composition: $F_{1,84} = 3.8$, $P = 0.061$). However, the length of the interaction that a fly had with another was strongly affected by group composition and fly state ((ANOVA)contact duration: group composition: $F_{1,84} = 14.8$, $P < 10^{-3}$; fly state: $F_{1,84} = 26.8$, $P < 10^{-3}$; fly state × group composition: $F_{1,84} = 22.9$, $P < 10^{-3}$). In homogeneous groups, cancerous flies had longer interactions compared to homogeneous control groups (Fig. 2b). However, when placed in a group of control flies (heterogeneous group), cancerous individuals showed a strong decrease in contact duration (Fig. 2b). Similarly, the average number of contacts per fly also differed depending on the social context and fly state ((ANOVA)number of contacts: group composition: $F_{1,84} = 17.5$, $P < 10^{-3}$; fly state: $F_{1,84} = 11.4$, $P = 0.001$; fly state × group composition: $F_{1,84} = 4.4$, $P = 0.038$). Groups of cancerous flies had a higher number of contacts than groups of control flies. Once again, cancerous individuals showed a decrease in the number of contacts when placed with control flies (heterogeneous group) compared to when in a group with other cancerous flies (Fig. 2c). Taken together this suggests that, individuals are more aggregated in a homogeneous group of cancerous flies than in a heterogeneous group or a homogeneous group of control flies. We thus concluded that, for a cancerous fly, the composition of the social group strongly affects the level of social interaction. However, our measure of social contact was constrained by the small size of the arena and therefore did not allow us to

disentangle the direction of the social contact i.e., whether specific fly states (cancerous or control) show avoidance or attraction towards other individuals within a group.

**Cancer progression and social environment choice**. Based on the results described above we tested whether cancerous and/or control flies would show variation in their choice of social environment and whether this was dependent on the level of their tumor progression. Using a similar protocol to Saltz[35], we assessed social preference by putting two small mesh cages, each containing 8 "stimulus flies" (cancerous or control) in a plastic, transparent box. The small mesh cages were placed on top of a small petri-dish containing standard food. We introduced a "focal fly" (cancerous or control) into the enclosed box and recorded their position over 7 h, i.e., whether the fly was found on one of the two mesh cages. Focal and stimulus flies were tested at different ages post HS-induction.

Cancerous flies appeared, on average, more attracted than control flies to other cancerous individuals and we observed a general decrease of preference by cancerous and control flies for the cancerous group with age of the focal fly (Fig. 3; focal fly: Wald $\chi^2_1 = 4.1$, $P = 0.04$; age: Wald $\chi^2_1 = 17.6$, $P < 10^{-3}$; age × focal fly: Wald $\chi^2_1 = 2.7$, $P = 0.1$). Furthermore, we find that at 21 days-post-HS, both control and cancer flies prefer to associate with control over cancer flies (Fig. 3).

To understand whether the preferences seen in the dual choice test, were due to avoidance or attraction, young (7 days post HS) focal flies were given a choice between a stimulus group in a mesh cage (8 flies) and an empty mesh cage using a similar experimental design. Cancerous flies showed, on average, attraction for the social group, independent of the age or state of the stimulus flies (Fig. 4a; intercept: Wald $\chi^2_1 = 8.1$, $P < 10^{-3}$; stimulus: Wald $\chi^2_1 = 0.06$, $P = 0.79$; stimulus age: Wald $\chi^2_1 = 1.4$, $P = 0.23$; stimulus × stimulus age: Wald $\chi^2_1 = 0.6$, $P = 0.44$). While control flies showed, on average, no clear attraction for the social group, they clearly avoided 3-week-old cancer flies (Fig. 4b; intercept: Wald $\chi^2_1 = 4.4$, $P = 0.036$; stimulus: Wald $\chi^2_1 = 2.6$, $P = 0.1$; stimulus age: Wald $\chi^2_1 = 3.37$, $P = 0.066$; stimulus × stimulus age: Wald $\chi^2_1 = 6.61$, $P = 0.01$).

**Discussion**
Here, we show that social environment can significantly shape the development of an intestinal-like cancer type in *Drosophila*. Consistent with previous studies on mammals[10,36], cancerous flies kept in isolation exhibit faster tumor progression than flies kept in groups of other cancerous individuals. However, more importantly we found that variation in group composition also leads to increased proliferation of tumor cells, thus highlighting how subtle variations in social structure may have dramatic effects on the progression of non-transmissible diseases. Other tumor models have been reported in adult *Drosophila*[37,38], which could be used in future projects to evaluate whether the social impact on tumor progression is a general or a tumor-specific effect.

Despite the opportunity to interact with others, individual cancerous flies, kept in groups with control flies, developed tumors at similar rates to when cancerous flies were bred in isolation. Social interaction analyses revealed that despite similar locomotor activities, cancerous flies interacted considerably less with control flies compared to when they were housed with other cancerous conspecifics. This reduction in social contact may potentially be perceived as a form of social isolation by cancerous flies, which could result in increased tumor progression, analogous to when flies are kept in true isolation. In humans, it has been proposed that the subjective perceived feeling of social

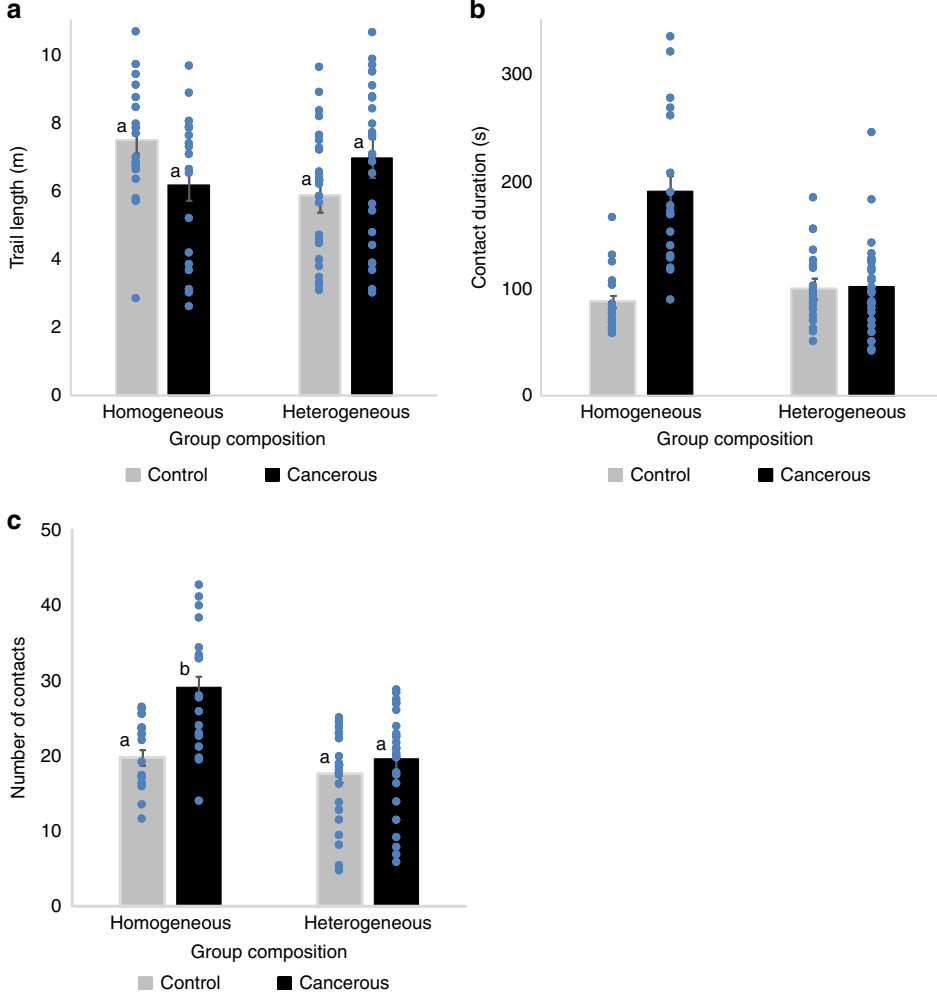

**Fig. 2** Social interactions for 21 day post-HS females in homogeneous (8 cancerous flies or 8 control flies) or heterogeneous groups (1 cancerous and 7 control flies). **a** Total locomotion trail. Statistical analyses were carried out on log trail length, but for simpler graphical presentation, we present the untransformed data. **b** The mean contact duration and **c** mean number of contacts an individual has, averaged across replicates. Blue dots indicate mean value for each replicate. Error bars: standard error of the mean. $N = 27$ heterogeneous groups and $N = 18$ homogeneous group for each fly state. Letters are Tukey's post-hoc classification

isolation may impact psychological and physiological traits as much as real social isolation[39]. Furthermore, the social environment choice experiment suggests that control flies may actively recognize and avoid cancerous individuals, especially when tumor progression is significant. Potentially, this may be a result of tumor-induced changes in cuticular hydrocarbons, pheromone profiles or even the gut microbiome. The most parsimonious explanation for this observation is infection avoidance[40,41], a behavior that has also recently been observed in *Drosophila*[42]. Even if not contagious, cancerous flies may show particular behaviors, or produce chemical cues, which are generally associated with being sick.

We find no such avoidance behavior in cancerous flies towards other cancerous individuals, suggesting that the benefits of slower cancer progression outweigh the costs of joining potentially infectious groups. Moreover, it also suggests that the fitness costs of joining sick individuals are lower than those of joining a group of healthy individuals. Firstly, with respect to predation risk, it is potentially worse to be the only sick individual within a healthy group than in a group with other vulnerable individuals (dilution effect). Secondly, since natural selection favors reproduction and not survival per se, and the probability to reproduce is reduced for ill individuals within a healthy group (because of avoidance of

sick partners and reduced competitive ability), it is better for sick individuals to join other ill individuals because sexual partners will be less selective and/or the disadvantage of sexual competition is reduced[43,44].

These findings offer new perspectives on the reciprocal relationship between disease and social behavior. While we observe that social structure has profound effects on disease progression, our study also suggests that disease might play a fundamental role in influencing group composition. We observed that cancerous flies, exhibit strong social attraction towards each other, especially at the beginning of tumor development, which decreases over time. At least two reasons could explain this change in preference. Firstly, at day 7 (when tumor progression is still minimal; Supplementary Figure 1a), it seems likely that flies are relatively unaffected by any pathological effects so that the decisions that maximize fitness related traits (i.e., slowing cancer progression and maximizing reproduction[19] can still be made, while at later stages of cancer progression (when tumor size is much more significant; Supplementary Figure 1c), the impact on normal functioning is likely to be high, and therefore the ability to make such decisions is lost. Secondly, it makes sense to be more selective during the initial stages of tumor progression, when the fitness benefits of reduced cancer progression are best maximized

(e.g., better reproductive output). In later stages, the "damage is done" and cancer will be too advanced to maximally reap these benefits. These findings raise questions on the very early impact of internal oncogenic process on individual behavior and natural selection pressures on this process[45]. While, at this stage, tumors are not found to affect the survival of flies, cancer may affect fitness in other ways (e.g., reproductive competitiveness, vulnerability to predators etc). A previous study showed that female *Drosophila*, bearing colorectal tumors, have earlier oviposition periods suggesting that flies are adapted to minimize the costs of cancer on fitness[19]. Further studies would be necessary to determine the exact proximate factors responsible for the effect found here, as well as the extent to which generalizations can be made across other cancer types (or indeed other illnesses) and animal species.

Our findings highlight the importance of social structure on disease progression, beyond the context of transmission. This is the first time that a direct link between social environment, specifically group composition, and cancer progression has been shown, while removing all other confounding psycho-sociological parameters that are frequently encountered in human studies. More generally, this study brings new light to how sociality impacts health and fitness in animals and its potential implication in human disease therapy. Moreover, we provide essential data for the emerging topic of evolutionary ecology of cancer, and demonstrate the importance of cancerous tumor progression in the intestine as a fitness-limiting factor that potentially influences life history adaptations and strategies.

## Methods

**Drosophila stocks and genetics.** *yw, HS-flp;esg-gal4, UAS-GFP;FRT82B,Tub-Gal80* (line 1), *yw,HS-flp;UAS-Ras$^{V12}$, FRT82B, Apc2$^{N175K}$, Apc$^{Q8}$* (line 2) and *yw, HS-flp, FRT82B* (line 3) flies[32] were balanced over co-segregating *SM5-TM6B* balancers. In all experiments, cancerous flies were *HS-flp;esg-gal4,UAS-GFP/UAS-Ras$^{V12}$;FRT82B,Tub-Gal80/FRT82B,Apc2$^{N175K}$,Apc$^{Q8}$* (offspring 1 of line 1 crossed to line 2), whereas controls were *HS-flp;esg-gal4,UAS-GFP;FRT82B,Tub-Gal80/FRT82B* (offspring 2 of line 1 crossed to line 3). MARCM clones[32] were randomly generated in heterozygous flies by flipase-induced exchange of pairing chromosome arms, resulting in mosaic individuals where homozygous *Apc2$^{N175K}$, Apc$^{Q8}$* mutant cells lacked the Gal80 repressor. This allowed Gal4 activity and the subsequent expression of GFP and Ras$^{V12}$ for clones located in intestinal progenitor cells. MARCM control clones are wild type for both Apc genes and do not express Ras$^{V12}$. MARCM clones were generated by heat shocking 3-day old female flies at 37 °C for 1 h[32]. Several attempts were made to use non-induced (no HS) offspring flies as controls, however, a few flies developed tumors without HS making this an inadequate control for our study.

**Flow cytometry.** These intestinal tumors are polyclonal and tumoral cells are often intercalated with healthy cells (Supplementary Figure 3), making it hard to delineate the limits of a tumor. For this reason we chose to quantify tumor size by flow cytometry (FACS) instead of measuring the area, which would only provide a rough estimate. Flies were starved overnight, provided only with water prior to the quantification of GFP-positive cells used to estimate tumor progression. The entire midgut and the Malpighian tubules (hereafter referred to generally as gut dissections), were sampled and dissected in PBS (phosphate buffer saline) as described[46]. Both tissues exhibited tumor-like structures. Each replicate consisted of the guts and Malpighian tubules of five flies. Each fly was taken from a separate tube of the same social environment treatment (for example, one replicate of the homogeneous treatment consisted of five guts and Malpighian tubules of cancerous flies, with each fly randomly taken from five different homogeneous tubes) and digested by collagenase (125 µg in 60 µl PBS) (Sigma-Aldrich) for 2 h at 27 °C with strong agitation. Sixty microliter Trypsin 10× (Sigma-Aldrich) was then added by gentle pipetting and nuclei were stained with Hoechst 33342 (0.5 µg/ml) for approximately 1 h. Dissociated cells were filtered through a cell strainer snap cap (70 µm size). To set the enzyme digestion protocol, we used Mex-gal4 > UAS-GFP (intestinal cells), esg-gal4 > UAS-GFP (stem cell and progenitors) and tumor-bearing guts, and checked the efficiency of the cell dissociation by direct

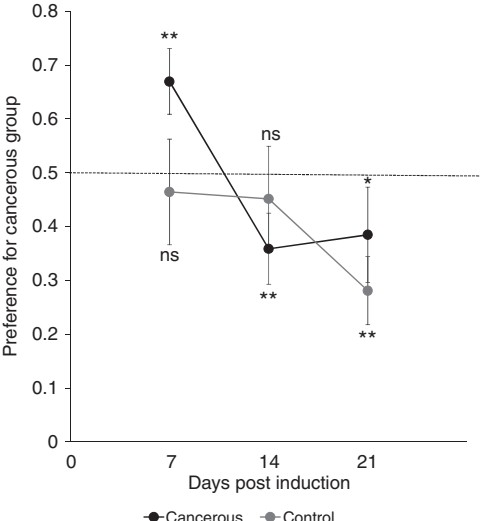

**Fig. 3** Dual choice experiment: proportion of lands on the mesh cage containing stimulus cancerous flies as a function of age. N = 12–21 per treatment. Stars indicate deviation from random choice (binomial test per state and age): ns: $P > 0.05$; *$P < 0.05$, **$P < 0.01$; Error bars: standard error of the mean

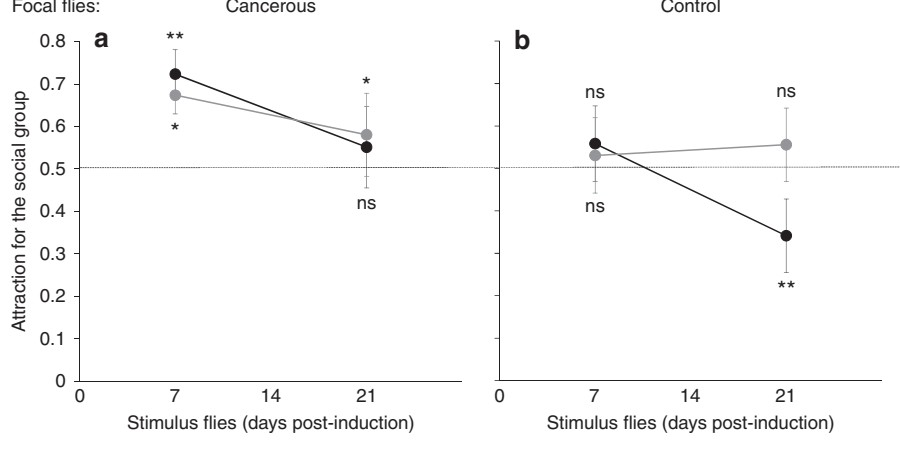

**Fig. 4** Attraction vs. aversion experiment. Proportion of lands on stimulus cage (vs. empty cage) by **a** focal cancerous flies and **b** focal control flies as a function of age of the stimulus flies. N = 16 per treatment. Stars indicate deviation from random choice (binomial test per state and age) ns: $P > 0.05$; *$P < 0.05$, **$P < 0.01$

observation with a GFP dissecting microscope every 30 min. The collagenase treatment was very efficient to digest the extracellular matrix, though the post-trypsin treatment was required to end up with single cells only. For each sample, 50,000 R2 cells were processed on a Partec PAS III and analyzed using the FlowMax software (Supplementary Figure 2, 3).

**Social environment**. Flies were sexed at emergence and control or cancerous females were kept in groups until the third day post emergence. Control and cancerous virgin females were heat shocked at 37 °C for 1 h. Flies were then put into their social groups and introduced into new 40 ml food tubes. Social groups created for experiments measuring tumor size consisted of either a cancerous fly in isolation, a group of eight all cancerous individuals (homogeneous groups) or a group of one cancerous individual in a group of seven non-cancerous individuals (heterogeneous group). Flies were partially wing-clipped on the right or left wing to distinguish their genotype. Previous behavioral studies have shown that wing clipping has no effect on social interactions[47]. Flies were then kept at 25 °C on standard food (changed every 3 days) until 21 days post-HS (induction). Flies were housed in small tubes (40 ml) to promote social interactions and limit the possibility of complete social isolation for any given fly. Tumor size was estimated with flow cytometry. Data (the number of tumor cells relative to the total number of cells counted) were analyzed using a generalized linear model (binomial distribution, Pearson correction for over-dispersion) and Tukey's post-hoc tests.

**Social interactions**. Again, females flies were put into groups according to the protocol described above, except this time social groups consisted of a group of eight all control flies (homogeneous), a group of eight all cancerous flies (homogeneous) and a single cancerous fly with seven control flies (heterogeneous). A group of flies to be tracked was composed of eight flies taken from different food tubes to ensure that they had never previously interacted. They were introduced into a semi-opaque white polyoxymethylene (Delrin) arena (diameter 100 mm; height 5 mm), covered with a transparent Plexiglas for 1 h. We simultaneously tracked four groups of eight flies over the 1 h. The tracking apparatus consisted of four synchronized firewire cameras (Guppy pro, Allied vision technologies), each filming one interaction arena that was backlit by a $150 \times 150$ mm IR backlight (R&D vision). We used Vision software to analyze spatial data (open-source C-trax 0.3.7[48]) that allowed us to collect ten positions per second for each fly[49,50] in the group over 1 h video experiments. Tracking corrections were made post C-trax analysis with fixerrors Matlab toolbox (Ctrax-allmatlab version 0.2.11) using Matlab software 7.11.0 to ensure that the identity of each fly was maintained when individuals were close to one another. For each fly, we calculated the total length of the path, the distance to other flies, the number of contacts with other flies (a contact was considered when the distance between the centers of two individuals was smaller than, or equal to, one mean body length of the individuals for 1 s or more) and the duration of each contact. Interactions were considered between all individuals within a group. We averaged each measure for all flies of the same cancer state within a group to obtain a single value for one replicate. For each measure we performed a general linear model and included the measures of group composition (homogeneous vs. heterogeneous), fly state (cancerous or control) and the interaction group composition × fly state as fixed explanatory variables. Tukey post-hoc contrasts among treatments were tested.

**Social environment choice**. Flies were sexed at emergence and control or cancerous females were kept in groups until the third day post emergence. Control and cancerous virgin females were heat shocked at 37 °C during 1 h and kept in groups until the day of the experiment. The experimental setup consisted of a $17 \times 12 \times 5$ cm plastic box in which 2 small $2 \times 2 \times 2$ cm mesh cages were introduced and each placed on a 3 cm diameter petri dish containing standard food. The two cages were positioned at opposite ends of the box. Groups of eight flies (hereafter referred to as stimulus flies) were placed in the mesh cages. In the dual choice experiment, one mesh cage contained control flies while the other contained cancerous flies. In the attraction vs. avoidance experiment only one of the two cages contained stimulus flies. A focal fly (control or cancerous), taken from a separate tube than the stimulus flies, was introduced in the box 15 h before starting the experiment. The position of the fly was then visually recorded every 30 min between 10 am and 5 pm. A choice was only recorded when the fly was positioned on a mesh cage or the associated petri dish. For the dual choice experiment, focal and stimulus flies were of the same age (7, 14, or 21 days post-induction), whereas for the attraction vs. aversion experiment the focal fly was always 7 days old post-induction and the stimulus flies were either 7 or 21 days post-induction. The number of times a focal fly was observed on a cancerous stimulus cage (for the dual choice experiment) or the stimulus cage (for the attraction vs. aversion experiment) compared to the total number of cage landings were then analyzed with a general linear model and a binary logistic regression. For the dual choice experiment, we first compared the behavior of cancerous vs. control flies: state of the focal fly was included as a fixed factor and fly age was included as a covariate. For the attraction vs. aversion experiment, we separately analyzed the behavior of each focal fly (i.e., cancerous or control) as a function of stimulus fly state and age. Finally, a binomial test, for each independent measure, was performed to test for a significant deviation from random choice.

## Data availability

The data and computer code uses to support the findings of this study are available from the corresponding author upon request.

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

## Acknowledgements

This work was supported by the ANR (Blanc project EVOCAN to F.T. and project DROSONET to F.M. and C.S.), the CNRS (INEE and INSB), *Fondation ARC* (1555286 to J.M. and F.M.), The French league against Cancer (M27218 to J.M.), IDEEV program (to F.M.), by an International Associated Laboratory Project France/Australia, by the French-Australian Science Innovation Collaboration Program Early Career Fellowship (B.U.), by André Hoffmann (Fondation MAVA), Fyssen Foundation (to F.M. and E.H. D.) and the French Government (fellowship 2015–155 to M.D.). We thank F. Bastin, J.C. Sandoz and A. Couto for their help with confocal imaging.

## Author contributions

E.H.D., T.B., C.M., F.T., J.M., and F.M. designed the experiments. E.H.D., T.B., J.D.S., C.M., M.S., B.M., and J.M. performed the experiments. C.S., J.M., and F.M. analyzed the data. E.H.D., J.M., F.T., A.C., C.S., B.U., and F.M. wrote the manuscript.

## Additional information

**Competing interests:** The authors declare no competing interests.

