## [Peer Review File · Nature Communications]

Reviewers' comments:

Reviewer #1 (Remarks to the Author):

This interesting paper analyzes the social behavior and cancer progression of fruit flies with genetically-induced colorectal-like tumors. The authors find that the flies' tumors actually progress most slowly when they are housed with other "cancerous" flies. Behavioral analysis indicated that the flies with cancer are ostracized from groups of healthy individuals, but have abundant social contact with other sick individuals, which may explain why these two types of groups have different effects. Finally, they show that flies with cancer prefer to associate with other sick flies.

These findings are very interesting because they are very surprising. The authors suggest that flies have evolved to prefer social groups that slow cancer progression, but at the same time the flies seem to show no other symptoms of the tumors for at least 3 weeks. The evolution of such a preference-performance correlation given these facts would be very surprising, because for such a correlation to evolve there must be strong and recurring selection. (e.g., see recent meta-analysis: <https://www.ncbi.nlm.nih.gov/pubmed/20100245>). If cancer has no major fitness effects for at least 1/3 of the flies' adult lives (in the lab) it is unclear how such strong selection could have occurred.

Similarly, Figure 3 indicates that cancerous flies initially prefer to associate with other cancerous individuals, but then change their minds as they get older. This is somewhat in conflict with the authors' interpretation that females associate with other cancerous individuals to reap fitness benefits of slowed cancer progression. Why would they only have this "adaptive" preference at day 7 but not at later days? Again, the phenomena are interesting, but hard to explain.

The authors suggest that many of the effects they observe are due to general sickness behavior, which makes sense. But, given the many known insect infections that are horizontally-transmitted, it is very surprising that any individual would prefer to associate with sick individuals. Even if there may be some benefits to slowing cancer progression, these may be outweighed by other large costs, such as the potential that the sick flies have something other than cancer (i.e., that is infectious and/or doesn't slow down your cancer), that predators might target groups of vulnerable individuals, etc.

These considerations don't make the findings of the paper incorrect, just very surprising. I think the authors should do a better job of highlighting that we really don't have a way of explaining these results currently, and that new theory may need to be developed (or, if I just missed the explanation, please clarify).

More minor points:

-The authors mention that their groups of healthy individuals were only females (line 127), but the cancerous (homogeneous) individuals are just described as "flies" (line 126). I assume these flies were also female, but this should be noted.

-I was very confused by figure 2 until I realized the data were aggregated at the level of the group, not the individual. Please clarify.

-I suggest that all papers include relevant statistical code as an appendix.

Reviewer #2 (Remarks to the Author):

Mery and colleagues present here an interesting study demonstrating reciprocal links between cancer progression and social interactions using a colorectal -like tumor model (Apc-Ras clones) in flies. While

the observations of this study are exciting and might open up new avenues of research in the field, there are a some technical concerns which need to be answered to validate the claims and strengthen the manuscript.

1. The authors use Apc-Ras clones for the entire study and generalize the results for all intestinal tumor models in flies, ignoring other models. To strengthen their claims, the authors should validate atleast some of the results in other classical fly tumor models which have been widely studied. For example tumors formed due to Notch inhibition in *esg+* cells (Ohlstein and Spradling, Nature (2006); Patel et al., NCB (2015)).

2. The authors use FACS for cell counting of MARCM clones. The gold-standard of quantifying MARCM remains staining and counting of cell/measuring area of the clones. While FACS sorting of GFP+ cells definitely is another way to quantify, the dissociation and filtering of cells/tumors before FACS could lead to loss of cells/tumor clumps which might skew the results. Therefore it is suggested that :

a. The authors should preferably include high resolution images of clones in the experiments and counts of the same.

b. Add more details for the FACS Sorting experiment for clarity, including total number of cells analyzed, number of living (propidium iodide (-) cells) etc. FACS sorting of *Drosophila* midgut cells has been performed by various labs, the papers/methods should be cited. Also, apart from taking the same number of guts, it is better to analyze the number of GFP+ cells out of the same number of input cells in the FACS. This is important as the guts of flies are not necessarily of the same healthy state and size with progressing age.

Reviewer #3 (Remarks to the Author):

In this manuscript, Dawson et al. perform experiments on *Drosophila* to show the impact of social environment on tumor progression, and how tumor progression, in turn, influences social interactions. The main conclusion of the manuscript is that cancerous flies kept in isolation or heterogeneous social environment show rapid tumor progression as compared to flies in homogeneous social groups. Finally, the study shows that flies have the ability to choose their social environment, with non cancerous flies actively avoiding flies at later stages of tumor development. I enjoyed reading the manuscript and think it makes a timely contribution to the literature. I have concerns that I would like to see better caveated in the manuscript (or addressed with changes to the analysis if feasible), and to see the thoughts of the authors on.

1) Social interaction experiment:

It is not clear from the method description whether contact duration/frequency is estimated in heterogeneous groups only when the contact occurs between a cancerous and a control fly. If instead, all contacts are considered, then the lines 146-147, 150-152 can be misleading. Since the heterogeneous group consists of seven control and only one cancerous fly (as opposed to eight control flies in the control homogeneous group), the social environment for control flies is very similar in the homogeneous and heterogeneous environment. Therefore, it seems quite obvious that the control flies demonstrated similar contact duration and frequency behavior in the two settings. Also, the comparisons made between control flies and cancerous flies does not seem fair because the heterogeneous group for control flies is quite socially homogeneous, which is not the case for cancerous flies.

There are two ways in which this point can be addressed: (i) by clarifying that only contacts between control and cancerous flies were considered in the heterogeneous group setting or (ii) providing a justification of why a social environment where most contacts presumably occur between control flies is considered to be heterogeneous for control flies.

2) Social environment choice experiment:

Line 168-171: Since the test is insignificant for age x focal fly interaction ($P=0.1$), there is no statistical evidence that there is "a general decrease of preference by cancerous flies for the cancerous group with age of the focal fly". Instead, the test results indicates that there is a general decrease of preference for the cancerous group with age for both control and cancerous flies ($P < 10^{-3}$).

3) The results of the dual choice suggest that 21 days post-induction cancerous flies prefer interacting with control flies over other 21 days post-induction cancerous flies (Figure 3). The attraction/revulsion experimental results further indicate that cancerous flies are not attracted towards 21 days post-induction cancerous flies (Figure 4A). These results seem contradictory to the observation of increased contact frequency/duration between cancerous flies in homogeneous groups (Figure 2). Can the authors provide an explanation towards this?

4) Discussion lines 190-195: The implications of this statement is that the social behavior of cancerous flies in heterogeneous environment is atypical. However, from figure 2, we see that the behavior of cancerous flies in heterogeneous group composition is similar to contact behavior of control flies. Instead, the contact behavior of cancerous flies in homogeneous environment appears to be atypical, with high rates of contacts than norm. Based on the results, I would therefore argue that cancerous flies demonstrate enhanced social interaction when housed together, rather than suggesting they experience "a form of social isolation" in heterogeneous social environment.

Minor comments:

Social environment experiment: The cumulative contact duration and contact frequency metrics can be highly correlated to each other. Have the authors corrected for such correlation? Instead of cumulative contact duration, average per-contact duration will be much more informative. Alternatively, only one of the two metrics can be presented in the results.

Results line 127-129 , Methods line 373-376: I am not sure how the reference group comprised of non-cancerous flies helps in this section which is about analysis of cancer progression in cancerous flies. Could the authors clarify?

Line 129-130: I would suggest careful choice of words. A statistically significant difference does not necessarily mean that the magnitude of the difference is dramatically high. The authors can either explicitly state what the difference (\pm SE) and provide justification why this difference is biologically high, or rephrase this statement.

Line 177-179: "control flies showed, on an average, no clear attraction for social group". I believe there is a mistake in this statement because the intercept is statistically significant. The significance of intercept was used in lines 174-176 to state that cancerous flies, in general, are attracted towards social groups.

Figure S1: The percentage of GFP+ positive cells in 21 day old cancerous flies is much lower than any of the three social scenario present in Figure 1. What generated this difference?

Figure 1 caption: "FACS analysis of GFP-positive in guts dissected from 21 days old control or

cancerous females" How are control flies represented in the figure? Please clarify.

Figure 2: Please also mention that the trail length is in log.

Figure 2 caption:

(a) Could you clarify the sample size of control flies and cancerous flies in heterogeneous groups? Since the ratio of flies is 1:7, the number of cancerous flies in heterogeneous group must be much less as compared to control flies.

(b) It seems that the heterogeneous group for the two fly state is not independent of each other because the recording was performed in the same arena (contrast this with the two homogeneous bars that represent recordings from independent arenas). The non-independence should be accounted in the general linear model by considering the arena id as a random effect.

(c) Delete "NS: P0.05, ***: $P < 10^{-3}$ ".

Reviewers' comments:

Reviewer #1 (Remarks to the Author):

This interesting paper analyzes the social behavior and cancer progression of fruit flies with genetically-induced colorectal-like tumors. The authors find that the flies' tumors actually progress most slowly when they are housed with other "cancerous" flies. Behavioral analysis indicated that the flies with cancer are ostracized from groups of healthy individuals, but have abundant social contact with other other sick individuals, which may explain why these two types of groups have different effects. Finally, they show that flies with cancer prefer to associate with other sick flies.

These findings are very interesting because they are very surprising. The authors suggest that flies have evolved to prefer social groups that slow cancer progression, but at the same time the flies seem to show no other symptoms of the tumors for at least 3 weeks. The evolution of such a preference-performance correlation given these facts would be very surprising, because for such a correlation to evolve there must be strong and recurring selection. (e.g., see recent meta-analysis: <https://www.ncbi.nlm.nih.gov/pubmed/20100245>). If cancer has no major fitness effects for at least

1/3 of the flies' adult lives (in the lab) it is unclear how such strong selection could have occurred.

This comment is very interesting and illustrates the current/classical view most ecologists and evolutionary biologists have on malignant pathologies. Recently, Thomas et al. 2017 & Thomas et al (in press) argue that although age is one of the strongest predictors of cancer and risk of death from cancer (i.e. cancer is therefore generally viewed as a senescence-related malady), cancer also exists at sub-clinical levels in humans and other animals, and its earlier effects on the body are poorly known by comparison. Oncogenic processes are a significant but ignored burden on the body and therefore likely to be a strong selective force from early in the lifetime of an organism. Although we didn't observe major fitness effect of cancer in the flies, we cannot exclude the possibility of malignant cell growths negatively impacting the physiology and health of the flies. For example, a recent study by Peck et al. (2016) showed that Tasmanian devils suffering from a transmissible facial tumour had significantly lower lymphocytes, erythrocytes, and haemoglobin concentration compared to healthy devils, a sign of acute phase response and inflammation, and anemia of chronic disease. Notably, no significant differences were found among stages of DFTD or ulcerated and nonulcerated tumors, indicating that cancer can negatively impact devil health even when the tumours are relatively small.

As mentioned in the introduction, cancer, like all diseases, is usually associated with trade-offs at some level, and at least for this reason the mechanisms employed by hosts to cope with cancer cannot be considered in isolation from other functions that govern living organisms. Moreover, recent work suggests that in addition to resistance mechanisms to cancer, selection has also favored adjustment of life-history traits, and tolerance mechanisms. Although metastatic cancers primarily cause major pathological manifestations at later life stages in laboratory animals, we should not underestimate the adaptation-invoking role of this disease in shaping the ecology and evolution of animals throughout the entire lifespan. So, while we found that survival in the laboratory was not influenced by tumour presence, we cannot rule out that tumor presence could have other indirect fitness burdens such as increased predator vulnerability of individuals or difficulties in finding mates, reproducing etc. We believe that this study supports the idea that it is time to adopt a novel view of malignant pathologies to improve our understanding of the ways in which oncogenic phenomena influence the ecology and evolution of animals long before their negative impacts become evident and fatal.

We have now made this clearer in the introduction (lines: 75-83)

Peck et al. 2015. Hematologic and serum biochemical changes associated with Devil facial tumor disease in Tasmanian Devils. *Veterinary Clinical Pathology*, 45: 417

Thomas et al. 2017. The importance of cancer cells for animal evolutionary ecology. *Nature Ecology and Evolution*, 1: 1592–1595.

Thomas et al. In press. Cancer is not (only) a senescence problem. *Trends in Cancer*.

Similarly, Figure 3 indicates that cancerous flies initially prefer to associate with other cancerous individuals, but then change their minds as they get older. This is somewhat in conflict with the authors' interpretation that females associate with other cancerous individuals to reap fitness benefits of slowed cancer progression. Why would they only have this "adaptive" preference at day 7 but not at later days? Again, the phenomena are interesting, but hard to explain.

At least two reasons can explain this contradicting decision pattern. First, at day 7 it seems plausible that the pathology has not yet significantly impaired normal fly functioning, so that decisions that maximise fitness related traits (i.e. slowing cancer progression and maximising reproduction before dying, e.g. Arnal et al. 2016) are still taken by flies; while later, due to the disease progression and its impact on the normal functioning, this capacity is lost. A second reason is more 'adaptive' in a state-dependent perspective: the fitness benefits of being choosy (to reduce cancer progression/reproduce) mostly exists when the cancer is initiated; once it is advanced, it is too late and fitness benefits cannot be maximised. Therefore, at least for these two reasons, it is predicted that cancer flies should be choosier at earlier stages of cancer progression than later. We have added this to the discussion (lines:218-227).

Arnal et al. 2016 Cancer brings forward oviposition in the fly *Drosophila melanogaster* Volume 7:272–276

The authors suggest that many of the effects they observe are due to general sickness behavior, which makes sense. But, given the many known insect infections that are horizontally-transmitted, it is very surprising that any individual would prefer to associate with sick individuals. Even if there may be some benefits to slowing cancer progression, these may be outweighed by other large costs, such as the potential that the sick flies have something other than cancer (i.e., that is infectious and/or doesn't slow down your cancer), that predators might target groups of vulnerable individuals, etc.

This is a very interesting and relevant comment. For this preference to evolve, we must indeed admit that the fitness costs of joining sick individuals are (despite contagion risks, predators etc...) lower than preferring a group of healthy individuals. This is, in our opinion, likely to be the case for the following reasons:

- In the face of predation risk, it is potentially worse (in term of predation probability) to be THE sick individual in a healthy group, than only one sick individual among other sick individuals (dilution effect).
- Concerning the infection risks, the additional costs of being infected are potentially small if the individual is already sick.
- Because natural selection favours reproduction and not survival per se, and that the probability to reproduce is probably slower for sick individuals in a healthy group (avoidance of parasitized partners, reduced competitiveness), it is better (in terms of mating opportunities) for a sick individual to join a group of sick individuals, because sexual partners will be less choosy and/or the disadvantage in sexual competition is reduced. This scenario has already been observed in other invertebrates infected by trematodes (e.g. Thomas et al. 1995, Campbell et al 2017).

Thomas et al. 1995, Assortative pairing in *Gammarus insensibilis* (Amphipoda) infected by a trematode parasite. *Oecologia* 995, Volume 104: 259–264

Campbell et al. 2017. An ecological role for assortative mating under infection? *Conservation Genetics* 18: 983-994

We now have added this to the discussion (lines: 204-213)

These considerations don't make the findings of the paper incorrect, just very surprising. I think the authors should do a better job of highlighting that we really don't have a way of explaining these results currently, and that new theory may need to be developed (or, if I just missed the explanation, please clarify).

In the light of the responses we provide above, we have now added several sentences in the text. We are grateful to referee 1 for having raised these important issues and we believe the reviewers' comments have made the manuscript stronger.

More minor points:

-The authors mention that their groups of healthy individuals were only females (line 127), but the cancerous (homogeneous) individuals are just described as "flies" (line 126). I assume these flies were also female, but this should be noted.

This has been corrected.

-I was very confused by figure 2 until I realized the data were aggregated at the level of the group, not the individual. Please clarify.

We have changed the legend to make this clearer and we have also clarified how we analysed this data in the text (Lines:428-433).

-I suggest that all papers include relevant statistical code as an appendix.

We carried out our analyses on SPSS. We have given a full description of all the statistical models in the text.

Reviewer #2 (Remarks to the Author):

Mery and colleagues present here an interesting study demonstrating reciprocal links between cancer progression and social interactions using a colorectal -like tumor model (Apc-Ras clones) in flies. While the observations of this study are exciting and might open up new avenues of research in the field, there are a some technical concerns which need to be answered to validate the claims and strengthen the manuscript.

1. The authors use Apc-Ras clones for the entire study and generalize the results for all intestinal tumor models in flies, ignoring other models. To strengthen their claims, the authors should validate at least some of the results in other classical fly tumor models which have been widely studied. For example tumors formed due to Notch inhibition in *esg+* cells (Ohlstein and Spradling, Nature (2006); Patel et al., NCB (2015)).

It is clearly very interesting to determine whether the findings here are a general or a tumor-specific effect and should indeed, be addressed not only in other *Drosophila* tumor models but also in other species. However, repeating the experiment with different cancer models would drive us far beyond the purpose and scope of this study. We do now however stipulate this in the discussion of the manuscript and make the caveat that our findings are potentially specific to this particular cancer model. Lines: 187-189.

2. The authors use FACS for cell counting of MARCM clones. The gold-standard of quantifying MARCM remains staining and counting of cell/measuring area of the clones. While FACS sorting of GFP+ cells definitely is another way to quantify, the dissociation and filtering of cells/tumors before FACS could lead to loss of cells/tumor clumps which might skew the results. Therefore it is suggested that :

a. The authors should preferably include high resolution images of clones in the experiments and counts of the same.

The way to evaluate tumor progression is obviously crucial, considering the purpose of our study. When starting the experiments, we searched for the best strategy to evaluate the net percentage of tumorous cells for each gut. Importantly, our preliminary experiments revealed that three weeks after heat shock induced recombination, the tumor cells appeared as condensed group of cells. We first considered measuring the size of these clones, but we noticed that the precise limit of tumors was difficult to delineate, since the tumor and surrounding control cells are at least, in part, imbricated. We have now added a high resolution image in the Supplementary Fig.1 to show this cell imbrication. In consequence, we estimated that measuring the percentage of GFP+ cells should be the most appropriate strategy to monitor tumor progression. To restrain cell loss, we worked to optimize the gut matrix digestion to produce single cells. We first observed that trypsin did not allow a full cell dissociation, but that collagenase digestion was very efficient, though a lot of doublets were still present at the end of the digestion. In contrast, we observed that the successive collagenase+trypsin treatment allowed the recovery of singles cells. This optimized protocol for gut digestion into single cells allows minimal cell loss and a very efficient strategy to measure tumor progression by FACS.

b. Add more details for the FACS Sorting experiment for clarity, including total number of cells analyzed, number of living (propidium iodide (-) cells) etc.

Measuring the number of dead versus living cells is an important issue to determine whether the difference in tumor progression results from changes in the rates of cell death or cell proliferation. In the course of our study, we dissected flies that had died the night prior to gut dissection, since we wanted to know whether these flies died due to giant tumors. Surprisingly, we could not find GFP+ cells in the abdomen of these flies, indicating that once dead, the GFP fluorescence is quickly lost. Therefore the rapid loss of GFP fluorescence does not permit us to measure the number of dead tumor cells by FACS analysis.

FACS sorting of *Drosophila* midgut cells has been performed by various labs, the papers/methods should be cited. Also, apart from taking the same number of guts, it is better to analyze the number of GFP+ cells out of the same number of input cells in the FACS. This is important as the guts of flies are not necessarily of the same healthy state and size with progressing age.

As mentioned above we worked to optimize the gut digestion protocol to minimize cell loss prior to FACS analysis. We now refer to a previous study describing the gut dissection+digestion protocol for FACS analysis and provide a more detailed description of our optimized protocol, including that we analyzed 50,000 R2 cells for each sample. ('Flow cytometry' section of material and methods)

Reviewer #3 (Remarks to the Author):

In this manuscript, Dawson et al. perform experiments on *Drosophila* to show the impact of social environment on tumor progression, and how tumor progression, in turn, influences social interactions. The main conclusion of the manuscript is that cancerous flies kept in isolation or heterogeneous social environment show rapid tumor progression as compared to flies in homogeneous social groups. Finally, the study shows that flies have the ability to choose their social environment, with non cancerous flies actively avoiding flies at later stages of tumor development. I enjoyed reading the manuscript and think it makes a timely contribution to the literature. I have concerns that I would like to see better caveated in the manuscript (or addressed with changes to the analysis if feasible), and to see the thoughts of the authors on.

1) Social interaction experiment:

It is not clear from the method description whether contact duration/frequency is estimated in heterogeneous groups only when the contact occurs between a cancerous and a control fly. If instead, all contacts are considered, then the lines 146-147, 150-152 can be misleading. Since the heterogeneous group consists of seven control and only one cancerous fly (as opposed to eight control flies in the control homogeneous group), the social environment for control flies is very similar in the homogeneous and heterogeneous environment. Therefore, it seems quite obvious that the control flies demonstrated similar contact duration and frequency behavior in the two settings. Also, the comparisons made between control flies and cancerous flies does not seem fair because the heterogeneous group for control flies is quite socially homogeneous, which is not the case for cancerous flies.

There are two ways in which this point can be addressed: (i) by clarifying that only contacts between control and cancerous flies were considered in the heterogeneous group setting or (ii) providing a justification of why a social environment where most contacts presumably occur between control flies is considered to be heterogeneous for control flies.

The contact duration/frequency for control flies in the heterogeneous group is estimated for interactions with all individuals within the group (control and cancerous included). The main purpose of this group (control flies within heterogeneous grp) is to act as a control to show the general level of social interactions of the heterogeneous environment (all interactions between control – control, control – cancer), which can then be directly compared to the cancer fly within this environment. Without this specific control it would not be possible to see how the cancer fly's behaviour compares to the general social behaviour of this particular environment.

Furthermore, since we cannot specify who initiates contact within an interaction (mentioned in lines: 156-158), the contact duration of cancer flies with control flies will be the mirror/same value for control flies with cancer flies. Therefore, it would not be informative to include this measure in the analyses. It is for this reason we carry out the social environment choice experiments so we can better gauge the social preferences of each fly state.

2) Social environment choice experiment:

Line 168-171: Since the test is insignificant for age x focal fly interaction ($P=0.1$), there is no statistical evidence that there is "a general decrease of preference by cancerous flies for the cancerous group with age of the focal fly". Instead, the test results indicates that there is a general decrease of preference for the cancerous group with age for both control and cancerous flies ($P < 10^{-3}$).

This was a mistake that has now been corrected.

3) The results of the dual choice suggest that 21 days post-induction cancerous flies prefer interacting with control flies over other 21 days post-induction cancerous flies (Figure 3). The attraction/revulsion experimental results further indicate that cancerous flies are not attracted towards 21 days post-induction cancerous flies (Figure 4A). These results seem contradictory to the observation of increased contact frequency/duration between cancerous flies in homogeneous groups (Figure 2). Can the authors provide an explanation towards this?

This is a good point, which highlights the comparability of these experiments. In the social interaction experiments (Fig 2), we evaluated the reciprocal choice of two flies to physically interact, whereas in the social choice experiments (Fig 3 & 4), we study the preference of individual flies to choose between social environments in which flies have no choice in the social interaction. Furthermore, the two experiments are not exactly comparable because in the social interaction experiment, flies have visual, touch and odour cues, whilst in the dual choice experiments they can only rely on odour cues. This could also explain the discrepancy in results.

4) Discussion lines 190-195: The implications of this statement is that the social behavior of cancerous flies in heterogeneous environment is atypical. However, from figure 2, we see that the behavior of cancerous flies in heterogeneous group composition is similar to contact behavior of control flies. Instead, the contact behavior of cancerous flies in homogeneous environment appears to be atypical, with high rates of contacts than norm. Based on the results, I would therefore argue that cancerous flies demonstrate enhanced social interaction when housed together, rather than suggesting they experience "a form of social isolation" in heterogeneous social environment.

We agree that this is one way to interpret the results of figure 2. However, we do not believe that our interpretation of "social isolation" is necessarily incorrect. The cancerous flies clearly seek out more social interactions (seen in the homogenous grp). So while this behaviour may not be the norm,

it does not negate the fact that the cancer flies are not receiving the necessary social contact, and hence could experience a form of social isolation. This is also backed up by the choice experiment, which shows that non-cancerous flies tend to avoid cancerous individuals.

Minor comments:

Social environment experiment: The cumulative contact duration and contact frequency metrics can be highly correlated to each other. Have the authors corrected for such correlation? Instead of cumulative contact duration, average per-contact duration will be much more informative. Alternatively, only one of the two metrics can be presented in the results.

Intuitively, it makes sense that contact duration and contact frequency are highly correlated. However, numerous papers (Kasper & Voelkl, 2009; Viblanc et al, 2016; Duboscq et al, 2016) show that even if these two parameters can be correlative they are not necessarily collinear. These two measures – duration and frequency – do not have the same meanings. The duration is more about the intensity of interactions, what we might call social activity, whilst the other is more about the quality of interactions, what we might call popularity. For instance, in Duboscq et al 2016, we showed that lice load is correlated with the number of contacts and not with the duration of them. Contacting many individuals on a short time may give more information than contacting one individual for a longer time.

For this reason we decided to keep both metrics in the manuscript.

Kasper, C., & Voelkl, B. (2009). A social network analysis of primate groups. *Primates*, 50(4), 343-356.
Viblanc, V. A., Pasquaretta, C., Sueur, C., Boonstra, R., & Dobson, F. S. (2016). Aggression in Columbian ground squirrels: relationships with age, kinship, energy allocation, and fitness. *Behavioral Ecology*, arw098.

Duboscq, J., Romano, V., Sueur, C., & MacIntosh, A. J. (2016). Network centrality and seasonality interact to predict lice load in a social primate. *Scientific reports*, 6, 22095.

Results line 127-129 , Methods line 373-376: I am not sure how the reference group comprised of non-cancerous flies helps in this section which is about analysis of cancer progression in cancerous flies. Could the authors clarify?

We have renamed and reworded this section as we agree it was misleading.

To clarify, for the main experiment, we only compared proportion of GFP+ cells of cancer flies, not control flies.

For preliminary experiments we did measure the GFP+ cells of control flies to show that the FACS strategy was appropriate and precise to measure cancer progression. As reported in the initial paper describing the tumor model (Martorel et al, 2014, PLoS One (92);e88413), the flies used as controls produce neutral clones that never associate in group of cells and are dispersed along the midgut at any time after heat shock-induced recombination.

There is also a misleading sentence in the legend of Figure 1 (which the reviewer also raises), which we have now also corrected.

Line 129-130: I would suggest careful choice of words. A statistically significant difference does not necessarily mean that the magnitude of the difference is dramatically high. The authors can either explicitly state what the difference (+ SE) and provide justification why this difference is biologically high, or rephrase this statement.

We agree that stats may not give a clear “magnitude” of the effect. The statistical test we did is based on 95% probability of difference and we do not assume any quantitative comparison. Based on the P value we were confident that there was a clear biological difference even if we could not quantitatively measure this.

Line 177-179: "control flies showed, on an average, no clear attraction for social group". I believe there is a mistake in this statement because the intercept is statistically significant. The significance of intercept was used in lines 174-176 to state that cancerous flies, in general, are attracted towards social groups.

This has now been corrected.

Figure S1: The percentage of GFP+ positive cells in 21 day old cancerous flies is much lower than any of the three social scenario present in Figure 1. What generated this difference?

The measurement of GFP+ cells as a function of age (now changed to figure S2) were done during our preliminary experiments. Following on from these preliminary experiments, we refined and improved the FACS settings, changing the GFP+ detection. This accounts for the difference in proportion of GFP+ cells in figure 1.

Figure 1 caption: "FACS analysis of GFP-positive in guts dissected from 21 days old control or cancerous females" How are control flies represented in the figure? Please clarify.

As mentioned above, there was a misleading sentence in the legend that we have now corrected.

Figure 2: Please also mention that the trail length is in log.

While we log transformed trail length to carry out statistical analysis, we graphically show untransformed trail length for easier visual interpretation. We have made this clearer in Figure 2 legend.

Figure 2 caption:

(a) Could you clarify the sample size of control flies and cancerous flies in heterogeneous groups? Since the ratio of flies is 1:7, the number of cancerous flies in heterogeneous group must be much less as compared to control flies.

For control flies in the heterogeneous groups, we averaged each parameter (e.g. contact duration etc) across the 7 control flies within the same group to obtain a single data point. Therefore, the sample size (N=27; written in Figure 2 legend) corresponds to 27 replicates for both cancer and control flies. This is also the case for the homogeneous groups, where an average for each group was obtained and used as a single data point. This is now mentioned in the text, lines: 432-433.

(b) It seems that the heterogeneous group for the two fly state is not independent of each other

because the recording was performed in the same arena (contrast this with the two homogeneous bars that represent recordings from independent arenas). The non-independence should be accounted in the general linear model by considering the arena id as a random effect.

Each individual was never tested twice. In this way, there is no pseudo-replication in our model. Furthermore, arena id cannot be considered as a random effect as group composition changed at each test. As each test corresponds to one arena Id, it is not informative to test it as a random effect.

(c) Delete "NS: P0.05, ***: $P < 10^{-3}$ ".

Deleted.

Reviewers' comments:

Reviewer #1 (Remarks to the Author):

The authors satisfactorily responded to the criticisms I raised in the first round of review.

Reviewer #2 (Remarks to the Author):

While the concept of the paper is quite interesting and the authors try to justify use of only FACS for measurement of tumours, I am still not convinced about the following points:

1) The authors state that they could not measure the MARCM clones as they were compact and in clumps. While I agree that the MARCM clones appear as condensed group of cells, it is still important to quantify the area and number of these clusters to corroborate their other results showing tumor progression. FACS is a good method to quantify, but a standard method like size measurement is required.

Further, while the authors mention that their FACS strategy is efficient for obtaining single cells, we cannot rule out the possibility that the undissociated clumps (tumors) might actually be filtered out when they use the cell strainer cap (what strainer size was used)? These are technical details which are very important to be considered specifically when the whole conclusion can be influenced by such technicalities. I still believe that addition of more experimental data by measurement of these MARCM clones is required to support the FACS results, in order to make the paper stronger and more credible.

2) Authors mention that they couldn't quantify living cells as there was no fluorescence as they flies were dead when dissected. I find the logic faulty and have the following concerns regarding this :

a) Why were flies dead for already a day used for FACS? This would lead to massive cell death of cells. Normally flies are anaesthetized and dissected just prior to FACS.

b) GFP signal is not the only way to access living cells. Use of propidium iodide separates live Vs dead cells.

Additional details and data are required to support the claims made by the authors.

Reviewer #3 (Remarks to the Author):

I enjoyed re-reviewing this manuscript which is much improved. The response of the authors to previous comments were well thought out and much appreciated.

I do have one more minor change to help improve the current version.

Line 40: "Flies kept in isolation exhibit faster tumor growth than flies kept in groups". This is not accurate because isolated flies exhibited faster tumor growth only compared to homogenous groups and not heterogeneous groups. I suggest revising this statement to "Flies kept in isolation exhibit faster tumor growth than flies kept in homogenous groups" or something similar.

Reviewers' comments:

Reviewer #1 (Remarks to the Author):

The authors satisfactorily responded to the criticisms I raised in the first round of review.

Reviewer #2 (Remarks to the Author):

While the concept of the paper is quite interesting and the authors try to justify use of only FACS for measurement of tumours, I am still not convinced about the following points:

1) The authors state that they could not measure the MARCM clones as they were compact and in clumps. While I agree that the MARCM clones appear as condensed group of cells, it is still important to quantify the area and number of these clusters to corroborate their other results showing tumor progression. FACS is a good method to quantify, but a standard method like size measurement is required.

After three years of experiments, we are convinced that the FACS strategy is the most efficient for the following reasons:

1) These tumors are polyclonal (A Casali, unpublished results), which means that a tumor is composed of more than one clone. Thus measuring the size of a given tumor does not reflect clone size, since we do not know how many clones constitute this tumor.

2) It is very difficult to precisely delineate the frontier of a tumor because of intricate healthy cells. Therefore, it would be inaccurate to measure the area of these intercalated tumors, which would only give us a rough estimate.

3) How should we consider small patches of GFP+ cells grouped in a restricted region of the gut (several examples are visible in the new Fig S3)? Since these structures are likely to evolve into a bigger, dense tumor a few days later (new Fig S3 K, L), should they be considered as several clones or one tumor? The FACS eliminates this uncertainty.

4) Since we are interested in subtle variations between populations, it is therefore necessary that we measure the *number* of GFP+ cells. The question being: how many tumor cells within a whole midgut? It is more robust to measure the *ratio* of GFP+ cells with respect to the total cell number in the host gut. We do not have the means by which to count the total number of cells within an entire gut except by FACS after single cell dissociation (see below for enzyme digestion efficiency).

5) Finally, a high ratio of GFP+ cells always corresponds to the appearance of a big dense tumor, and a low ratio corresponds to dispersed tiny clones/tumors. Prior to enzyme digestion, we always visually inspected the dissected guts under the microscope to ensure that the FACS results were correct. It would be impossible to show a correlation between measurements obtained through confocal imaging and the FACS since the gut samples would be destroyed and not recoverable for enzyme digestion for FACS analysis.

To support our statement, we are providing a new supplementary figure (S3), where we dissected twelve guts, imaged them in PBS immediately after dissection with a GFP dissecting microscope prior to enzyme digestion and FACS analysis. The images show a strong correlation between the pictures and the ratio of GFP+ cells measured by FACS.

Moreover, to highlight the points we make above, we would like to draw attention to pictures J (which appears very dense) and picture G, which too appears large but is in fact intercalated with many healthy cells. The FACS measurements obtained for these two samples reflect these differences emphasizing the validity of the FACS protocol rather than a rough evaluation of clone/polyclonal group size to precisely monitor tumor progression.

Further, while the authors mention that their FACS strategy is efficient for obtaining single cells, we cannot rule out the possibility that the undissociated clumps (tumors) might actually be filtered out when they use the cell strainer cap (what strainer size was used)? These are technical details which are very important to be considered specifically when the whole conclusion can be influenced by such technicalities. I still believe that addition of more experimental data by measurement of these MARCM clones is required to support the FACS results, in order to make the paper stronger and more credible.

Before starting the experiments, we were extremely concerned about the best strategy to measure tumor progression. Therefore, we carried out many preliminary tests to find the best protocol. We used *esg-gal4>UAS-GFP* (stem cells and precursors), *mex-gal4>UAS-GFP* (enterocytes) and GFP-tumor bearing guts. We tested collagenase or trypsin, or a mix for various times (1h, 2h, 4h, overnight) at different temperatures (4C, 18C, 25C, 27C) with mild rotating agitation. We checked the efficiency of cell dissociation by direct observation with a GFP dissecting microscope every 30 minutes. We noticed that after 2h at 27C with collagenase, the intestinal cells were still together, but that the tumor cells tended to be separated. We were also surprised to observe that the peritrophic membrane tended to maintain the gut cells together and that tumor cells were easier to dissociate than the gut cells (actually, big tumors tend to weaken the gut integrity). Nonetheless, gentle pipetting was sufficient to dissociate the gut, although a few cells still remained as clusters. We then added trypsin and observed a complete dissociation to single cells (trypsin alone, without the previous collagenase treatment was very inefficient).

Therefore, the final digestive protocol consisted of collagenase treatment for 2h at 27C in 1,5ml Eppendorf tubes under a strong agitation on a thermomixer. This was followed by the addition of trypsin+Hoechst through gently pipetting the sample, to help dissociation. After 30mins at 27C under a strong agitation, we observed a complete dissociation to single cells. Since we observed that our protocol led to a complete dissociation to single cells, we are confident that our strainer cell cap (70 μ m) did not filter out undissociated tumors.

We have added more details of this protocol to the methods section of the manuscript.

2) Authors mention that they couldn't quantify living cells as there was no fluorescence as they flies were dead when dissected. I find the logic faulty and have the following concerns regarding this :

a) Why were flies dead for already a day used for FACS? This would lead to massive cell death of cells. Normally flies are anaesthetized and dissected just prior to FACS.

b) GFP signal is not the only way to access living cells. Use of propidium iodide separates live Vs dead cells.

We apologize for the unclear statement in our previous response. None of the FACS experiments used in the manuscript were performed with dead flies. All flies were alive and anaesthetized prior to being dissected.

Our previous response referred to preliminary investigations into whether flies were dying because of big tumors (which we have not included in the manuscript). We dissected some of the flies that died during the night of starvation prior to gut dissection and FACS analysis. We were not able to see any tumors (big or small) in these dead flies, which suggests that the GFP is quickly lost in dead tumor cells. The question of whether tumor progression depends on a change in cell death and/or proliferation rate would involve uncovering the underlying physiological mechanisms, which drives us far beyond the scope and purpose of this study. We plan to investigate this issue in the future, but first, we will have to perform many preliminary experiments for setting the optimal strategy to address this issue. We have modified the text of the manuscript to read "tumor progression" rather than "tumor growth", since growth suggests an effect on proliferation rate.

Additional details and data are required to support the claims made by the authors.

Reviewer #3 (Remarks to the Author):

I enjoyed re-reviewing this manuscript which is much improved. The response of the authors to previous comments were well thought out and much appreciated.

I do have one more minor change to help improve the current version.

Line 40: "Flies kept in isolation exhibit faster tumor growth than flies kept in groups". This is not accurate because isolated flies exhibited faster tumor growth only compared to homogenous groups and not heterogeneous groups. I suggest revising this statement to "Flies kept in isolation exhibit faster tumor growth than flies kept in homogenous groups" or something similar.

This has now been corrected.

REVIEWERS' COMMENTS:

Reviewer #2 (Remarks to the Author):

The authors have answered most of my concerns in this round of revision and have made suitable corrections where required.

Addition of Supplementary S3 is important and I suggest that the rationale behind using FACS (as authors have described in the rebuttal letter) be briefly discussed in the manuscript for the better understanding of readers.